# An Innovative Method of Improving an Extract of *Andrographis paniculata* from Leaves: Its Anticancer Effect Involves the Cell Endoplasmic Reticulum

**DOI:** 10.3390/ijms26010344

**Published:** 2025-01-02

**Authors:** Jessica Maiuolo, Rosamaria Caminiti, Valeria Mazza, Francesca Oppedisano, Saverio Nucera, Maria Serra, Roberta Macrì, Ernesto Palma, Annie Eng, Vincenzo Mollace, Carolina Muscoli

**Affiliations:** 1Department of Health Sciences, Institute of Research for Food Safety and Health (IRC-FSH), University “Magna Graecia” of Catanzaro, 88100 Catanzaro, Italy; maiuolo@unicz.it (J.M.); r.caminiti@unicz.it (R.C.); valeria.mazza@unicz.it (V.M.); saverio.nucera@hotmail.it (S.N.); maria.serra@studenti.unicz.it (M.S.); robertamacri85@gmail.com (R.M.); palma@unicz.it (E.P.); mollace@libero.it (V.M.); muscoli@unicz.it (C.M.); 2HP Ingredients North Port, Bradenton, FL 34205, USA; annie@hpingredients.com

**Keywords:** *Andrographis paniculata*, andrographolide, endoplasmic reticulum, UPR, apoptosis, anticancer role, phytotherapeutic effect

## Abstract

In this manuscript, the effects of two extracts from *Andrographis paniculata* were tested: (a) an extract titrated to 49.7% of andrographolide and obtained from leaves of the plant: (b) the pure andrographolide titrated to 99%. The extracts were dissolved in 1-butanol and tested on tumor lines (MCF7 and SH-SY5Y) and the non-tumor line (Huvec) to understand the effects on cell proliferation. The addition of a sonication process improved their dissolution and efficacy making these extracts unique and innovative. The experiments conducted (viability measurements, solubility of the extracts, IC_50_ tests, measurement of oxidative potential, lipid and cytosolic calcium concentration, and mortality assessment by annexin assay) showed a different behavior of the extracts on cancer cells and not. In particular, the extracts did not cause toxic effects on the viability of the Huvec cells, while both tumor lines were damaged, demonstrating that cancer cells are more susceptible to extracts of *A. paniculata* than healthy cells. The mechanism of action responsible for the damage detected involved the functioning of the endoplasmic reticulum organelle and finally resulted in apoptotic death. For this reason, the extracts considered have shown a potential anti-tumor role and *A. paniculata* could be used and exploited in pharmacological therapy against cancer. However, further studies, obtained in clinical practice, should be conducted to increase knowledge of the effects of *A. paniculata* on the organism and its phytotherapeutic role.

## 1. Introduction

*Andrographis paniculata* (Burm. F.) Nees (*A. paniculata*), known as the “king of bitters”, is an annual herbaceous plant belonging to the family of Acanthaceae and the genus *Andrographis*. * A. paniculate* is native to Southeast Asia and appears mainly in India and Sri Lanka. However, it is also present in many tropical and subtropical countries, including Vietnam, Malaysia, Thailand, and Mauritius [1,2]. The phytotherapeutic potential of *A. paniculata* has already been recognized by Indian and traditional Chinese medicine [3]. Today, it is considered for its antipyretic, anti-inflammatory, anticancer, antiviral, antithrombotic, immunostimulatory, hypoglycemic, and hypotensive activities [4]. These biological properties are justified by the composition of the extract, including flavonoids and diterpenes among others. The main flavonoids were apigenin, 7-O-methylwogonin, 3,4-dicarboxyylquinic, and onysylin [5]. Diterpenes are composed of four isoprene units (C_5_H_8_) which are easily oxidized to develop one or more lactone groups, with different levels of solubility [6]. *A. paniculata* consists of several diterpene lactones with different bioavailability among which the main ones are the andrographolide, 14-Deoxy-11,12-didehydroandrographolide, neo-andrographolide, and 14-deoxyandrographolide [7]. Among these, andrographolide (C_20_H_30_O_5_) is responsible for anti-inflammatory, anticancer, anti-central nervous system dysfunctions, and anti-platelet properties. However, its low aqueous solubility is responsible for poor availability following oral administration [8,9]. Recently, the cytotoxic effects of andrographolide have confirmed its potential use against the uncontrolled growth typical of cancer [10]. Moreover, these effects seem to be specific to certain cancer lines with mechanisms that vary depending on the types of cancer cells. This aspect could be interesting in the pharmacological treatment of some types of cancer [11].

The endoplasmic reticulum (ER) is a cellular organelle consisting of several structural domains whose main functions are protein synthesis, transport, and folding, lipid synthesis, calcium ion storage, and carbohydrate metabolism [12]. The shape, size, and function of the ER are highly dynamic and reflect changes within the cell due to various factors such as cellular differentiation, development, intracellular signals, and cell cycle stages, among others. Numerous environmental, genetic, or toxic insults are responsible for altering the ability of cells to form functional proteins. These alterations involve the functioning of the endoplasmic reticulum, which is no longer able to fold or modify the proteins post-translationally, leading to an accumulation of misfolded proteins [13]. Cells stressed by the endoplasmic reticulum must quickly restore the physiological functions of the organelle and activate an intracellular signaling pathway called the “Unfolded Protein Response” (UPR) [14]. UPR reduces protein synthesis, increases cellular “folding” capacity, and induces cytoprotective genes [15]. Three transmembrane transducers trigger it: (a) inositol-requiring enzyme 1α (IRE1α), (b) pancreatic endoplasmic reticulum kinase (PERK), and (c) activating transcription factor 6 (ATF6) which by activating, individually or sequentially, can switch off the stress of ER via different pathways [14]. In particular, PERK is a transmembrane protein of ER which, when activated, is oligomerized and trans-autophosphorylated, activating its kinase domain and phosphorylating its main target eIF2α, a starting factor of protein synthesis. eIF2α terminates the translation of most mRNA, including transcription factor 4 (ATF4). ATF4 promotes amino acid biosynthesis and expression of antioxidant responses and transports genes to maintain cell survival [16]. In addition, ATF4 controls the expression of many adaptive genes facilitating cell survival even under prolonged stress conditions. However, if the period of cellular stress is too prolonged or in cancer cells, where ATF4 is over-expressed, this transcription factor can promote apoptotic death [17]. In recent years, a correlation has been shown between andrographolide and ER stress, although the mechanisms involved are variable according to the experimental model chosen [18,19].

In this manuscript, the effects of an extract of *A. paniculata,* obtained from the leaves of the plant and titrated to 49.7% andrographolide (ALE), will be tested and compared with another extract from the same plant but titrated to 99% andrographolide (PA). This comparison will offer important conclusions on the effects of andrographolide, which is always present but in different concentrations. The following will also be evaluated:
(1)The use of different alcohol solvents, which vary according to their hydrocarbon group length, to dissolve ALE and PA.(2)The improvement of the effectiveness of ALE and PA by varying some dissolution parameters.(3)The effect of ALE and PA on cancer cell lines to assess their potential role against cancer; for this purpose, a human neuroblastoma line (SH-SY5Y), a human breast cancer line (MCF-7), and a non-tumor line of human umbilical vein endothelial cells (Huvec), used as the control line, will be analyzed.

## 2. Results and Discussion

### 2.1. Effects of ALE and PA Dissolved in Alcoholic Solvents

ALE and PA were solubilized in four different solvents: methanol, ethanol, propanol, and butanol. These solvents are identified as “alcohols” due to the presence of the hydroxyl group (-OH) and differ in the number of carbons (methanol = 1, ethanol = 2, propanol = 3, and butanol = 4). Alcohols are derived from alkanes by substitution of a hydrogen atom with an -OH hydroxyl group. These solvents are considered “short chain” and for this reason appear liquid; their good miscibility with water is justified by the presence of the -OH group, which can form hydrogen bonds. However, the solubility decreases with increasing number of carbon atoms in the hydrocarbon chain, as hydrophobic characteristics prevail: the increase in the number of groups -CH_2_ reduces proportionally the solubility in water and increases the affinity to the apolar compounds [20]. The choice of these solvents is justified by the desire to maintain high hydrophilicity compared with the countless organic solvents. Several concentrations of ALE and PA were tested (1.5–100 µg/mL) on the three cell lines, and the results obtained showed that (Figure 1):
(a)when the compounds were dissolved in 1-butanol, they exhibited significantly higher toxicity than all the other solvents.(b)ALE extract (in any solvent) has always been more toxic than PA.(c)In non-cancer cells, only the highest concentrations of ALE and PA exhibited toxicity of no more than 30%. On the contrary, cancer lines reported significantly higher toxicity following treatment with ALE and PA.(d)Neurons were found to be more sensitive than breast cancer cells, and in fact, ALE and PA were more toxic.

**Figure 1 ijms-26-00344-f001:**
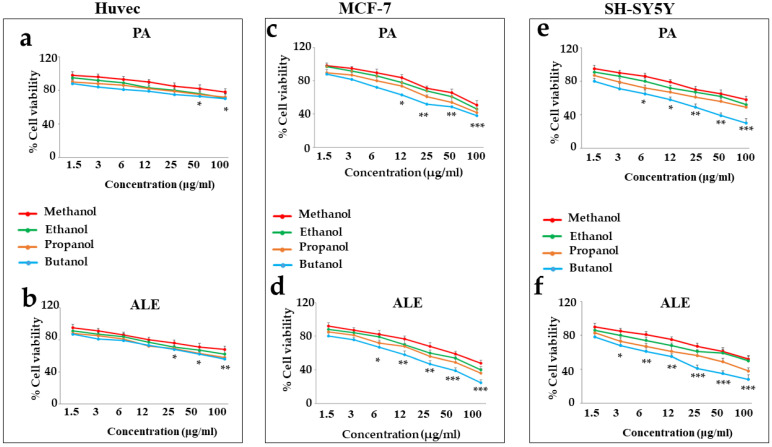
Effects of PA and ALE on cell viability. Figure 1 shows the effect of ALE and PA, dissolved in several solvents, on the three cell lines chosen: Huvec panels (**a**,**b**), MCF-7 panels (**c**,**d**), and SH-SY5Y panels (**e**,**f**). Three independent experiments were carried out, and the values are expressed as the mean ± SD. * denotes *p* < 0.05 vs. the lowest concentration ** denotes *p* < 0.01 vs. the lowest concentration; *** denotes *p* < 0.001 vs. the lowest concentration; A Tukey–Kramer comparison test followed the analysis of Variance (ANOVA).

The results obtained can be explained by the knowledge that ALE contains the compound andrographolide (49.7%), which is characterized by an apolar portion and is, therefore, more miscible to butanol [21]. The toxic effect could be precisely attributed to the diterpenoid andrographolide, which has shown anticancer potential in the development and progression of different types of cancer [22]. If andrographolide is responsible for the observed toxicity, we expect that PA is more toxic than ALE, since it contains a higher amount (98%) than ALE. This apparent discrepancy could be explained by the poor solubility of andrographolide: when its percentage is lower (ALE), butanol could ensure total dissolution. In contrast, andrographolide in PA may not dissolve completely in butanol, leaving microscopic traces of an “unbroken” substance [23].

### 2.2. Comparison of IC_50_ of ALE and PA Dissolved in Several Solvents

The results obtained in these experiments showed that when ALE and PA were dissolved in 1-butanol, they showed a lower IC_50_ than all others, demonstrating that a lower concentration of these products is required to induce 50% mortality (Figure 2). This effect has already been evident in Huvec cells, although it appeared more pronounced in cancer cell lines. In Huvec cells, IC_50_ was 103.71 and 86.1 µg/mL for PA and ALE, respectively. The reduction in ALE’s IC_50_ compared to PA’s was statistically significant. In MCF-7 cells, the IC_50_ was inferior to non-cancer cells (28 and 21 µg/mL for PA and ALE, respectively), showing that the cancer cells need a lower amount of extracts to reduce viability by 50%. In addition, the difference between PA and ALE is negligible and not significant. Finally, neuroblastoma cells appeared more sensitive, with IC_50_ values even lower (26 and 10 µg/mL for PA and ALE, respectively). The difference observed was also significant in this situation. Considering the results obtained, the higher toxicity demonstrated by PA and ALE when dissolved in 1-butanol, their potential anti-tumoral effect, and the lower IC_50_ value evidenced in this solvent led us to choose 1-butanol from among the solvents considered. It is important to know that the cellular toxicity induced by PA and ALE is not due to the vehicle in which they are dissolved, as is demonstrated and represented in Appendix A.

In addition, the dose–response and time–response curves following treatment with both extracts are shown in Appendix A. In Appendix A, the effects of PA and ALE can be observed at increasing concentrations (0–100 μg/mL). In Huvec cells, the two curves are superimposed while in MCF-7 and SH-SY5Y they are proportionally separated, with ALE causing greater damage than PA. In Appendix A, the effects of PA and ALE over time were measured. As shown, the greatest effect is observed at 24 h, while at longer times (up to 72 h), the effects froze.

### 2.3. Sonication Improves the Anticancer Effects of the Extracts

Since the poor solubility and dissolution rate of andrographolide is known [24], ALE and PA after being dissolved in 1-butanol have been subjected to a process of sonication, which could intensify and improve the dissolution. In general, ultrasonic homogenization depends on sonication time and power [25], and the numerous tests, carried out in our experimental model, are shown in Figure 3. In Figure 3a, the results obtained from the sonication of ALE and PA for several lengths of time (0, 1, 5, 10, 15, and 20 min) are reported. As shown, sonication improved the anti-tumoral effect of both ALE and PA compared to non-sonicated samples, and cell viability was significantly and gradually reduced because of this process. The time of better effect was 10′ of sonication, while the later times showed similar effects. In Figure 3b, differences in cell viability, obtained before and after sonication for 10′, are summarized. The increase in effects obtained after sonication can be explained by changes in the size, morphology, and degree of crystallinity of the particles contained in the extracts [26]. In addition, to prevent a possible breakdown of the chemical components of the extracts, due to overheating during sonication, samples were always kept in an ice-containing container. The efficiency of sonication in improving extracts had already been demonstrated in a previous paper obtained and published by our research group [27]. ALE and PA have been sonicated for 10′ in the continuation of the experimental work.

### 2.4. Antioxidant Potential of PA and ALE

Andrographolide showed a changing effect on oxidative stress, demonstrating an antioxidant and sometimes pro-oxidant role [28,29]. This contradictory behavior can easily be justified by several factors, such as the concentration of the compound: in general, high concentrations promote a pro-oxidant effect, while lower concentrations tend to provide antioxidant properties [30,31]. In our study, we evaluated the oxidative potential of PA and ALE both directly on the extracts and on the cell lines considered, with the results shown in Figure 4 and Figure 5, respectively. Figure 4 shows the antioxidant activity of the extracts in vitro, measured by the ORAC test (Figure 4a), their chelating activity (Figure 4b), and the measurement of their reducing potential (Figure 4c). As can be seen from the results, PA and ALE showed a modest antioxidant property (Figure 4a, demonstrated by the position of the corresponding curves, placed between the curve of Trolox 7.6 μg/mL and the negative control) and a more pronounced chelating and reducing properties at 10 μg/mL (Figure 4b,c).

Figure 5 shows the formation and accumulation of reactive oxygen species (ROS) in cell lines. In Huvec cells, PA and ALE did not produce ROS, as can be seen from their values which are similar to untreated cells (Figure 5a). In contrast, in MCF-7 and SH-SY5Y cells, the extracts performed pro-oxidant activity significantly increasing the accumulation of ROS compared to the control (Figure 5b,c). This effect appeared greater in the neurons than in MCF-7, demonstrating, once again, their delicacy and sensitivity. In addition, ALE has been shown to have a statistically higher pro-oxidant effect than PA, increasing the accumulation of ROS in cancer lines. The conflicting results obtained with cells cannot be justified by very high concentrations: in vitro experiments and on cell lines, the same concentration of PA and ALE was used (10 μg/mL). One explanation for this phenomenon is provided by some physiological differences between healthy and cancer cells. One of these could be the intracellular pH, which is physiologically maintained within a narrow range. The pH homeostasis regulates many cellular functions, such as growth, migration, and apoptotic death. Cancer cells may exhibit altered pH values resulting in inhibition of migration, avoidance of apoptosis, and proliferative progression [32]. Altered pH or similar in cancer cells may explain the different effects of extracts compared to non-cancer cells [33].

### 2.5. Involvement of Endoplasmic Reticulum

The cellular lipid component plays fundamental roles in cells, such as the composition of biological membranes, signaling activity, regulation, and energy source. For this reason, a dysregulation of the lipid balance could facilitate cell dysfunction. Today, numerous literature studies have shown that cancer cells are made up of high alterations in lipid composition to support rapid proliferation, and lipid catabolism ensures a continuous supply of ATP, necessary for the proliferation of cancer cells [34]. Consequently, the compounds that alter the lipid content could be considered therapeutic anticancer agents [35]. With this in mind, we decided to test the effects of ALE and PA on lipid and intracellular calcium concentrations, two characteristics governed and physiologically regulated by the smooth endoplasmic reticulum. The results are shown in Figure 6. Once again, the cancer cell lines had a different effect than the Huvec: the treatment with ALE and PA resulted in a significant reduction in the lipid component in MCF-7 and SH-SY5Y. Neurons were more responsive than MCF-7, and in these cells, ALE was statistically more effective than PA in reducing lipid concentration (Figure 6a). Figure 6b shows changes in cytosolic calcium concentration following treatment of the selected cell lines with ALE and PA. Huvec cells are not affected by the treatment with extracts and the cytosolic calcium profile is identical to untreated cells. A similar argument cannot be made for the tumor lines, in which the cytosolic calcium is reduced by ALE and PA. The calcium ion is a highly important cell molecule that controls a wide variety of processes. Between these, the proliferation, migration, cell death, and immune response are also critical to the initiation and progression of cancer. The results obtained were very interesting since the compounds that cause calcium dysregulation could be a therapeutic basis for cancer control [36].

### 2.6. PA and ALE Involve PERK Sensor of UPR

The involvement of the endoplasmic reticulum led us to search for the corresponding pathway, so we can know the specific mechanism of action. For this reason, we decided to study the UPR. In our experimental model, only the PERK branch of the UPR was involved (Figure 7 and Figure 8), while the pathways connected to the IRE-1 and ATF-6 sensors showed no activation. The activation of UPR, involving the PERK sensor, is characterized by increased expression of its phosphorylated form and phosphorylation of the protein eIF2α, the essential factor in initiating eukaryotic protein synthesis. As can be seen in Figure 7, the treatment of tumor lines with ALE and PA led to a significant increase in the expression of these proteins, while Huvec cells showed their expression like or just above untreated cells. The respective quantifications are shown in the lower part of the figure. Signaling of activated PERK was followed downstream of eIF2α, with the quantification of the expression of ATF4 (Activating Transcription Factor 4), a transcription factor with the ability to form dimers with many different proteins that affect gene expression and cellular fate. Figure 8 shows the expression of ATF4, evaluated by cytofluorimetric readings. The results highlighted a similar pattern to those discussed above, and only the cancer cell lines showed an increased expression of this transcription factor. The activation of ATF4 is related to the decision to lean towards survival or cell death (apoptosis) [37]. It has been shown that the tumor micro-environment (reduced nutrients, hypoxia, and altered pH) can affect the functioning of the endoplasmic reticulum in maintaining cellular homeostasis; therefore, a close correlation exists between cancer and endoplasmic reticulum [38]. To date, it has been found that natural products and their derivatives have anticancer effects by inducing organelle stress [39]. In this direction, we can interpret the results obtained as an anticancer strategy.

### 2.7. UPR Activation, Induced by PA and ALE, Promotes Cell Apoptotic Death

The damage generated by treatment with ALE and PA, in tumor cell lines, resulted in the dysfunction of the endoplasmic reticulum and culminated in the activation of apoptosis, as shown in Figure 9. On the contrary, the non-tumoral Huvec cells did not generate apoptosis following treatment with the extracts. We can conclude that ALE and PA induced stress in the endoplasmic reticulum and activated the UPR, and trinomial PERK-eIF2α-ATF4 could prolong the stress of the endoplasmic reticulum until the apoptotic pathway is triggered [40].

## 3. Materials and Methods

### 3.1. Plant Material

To obtain ALE extract, the leaves of *A. paniculata* were harvested in India in April 2021 and subjected to alcoholic extraction (75% ethanol); the ratio of herbal drug to extract was 10:1. Following the botanical recognition, the voucher specimen n° 10112021-006 was awarded and kept at HPIngredients, FL, USA. The composition of ALE was analyzed by high-performance liquid chromatography (HPLC) after the compounds were extracted with acetone (4:1). An inverse phase LiChrospher RP C18 column (4125 mm; Sigma Aldrich, Santiago, Chile) was used; the mobile phase consisted of 26% acetonitrile and 0.5% phosphoric acid, eluted at a rate of 1.1 mL/min using a wavelength of 228 nm. The following reference standards were used: andrographolide (purity > 98%), 14 deoxyandrographolide (purity > 90%), and neoandrographolide (purity > 90%). The results obtained showed:
(1)andrographolide 49.7%(2)14 deoxyandrographolide 0.2%(3)neoandrographolide 0.2%

At the end of the extraction procedure, ALE was registered as PAR-210501-IN (HPIngredients, Bradenton, FL, USA).

### 3.2. Cell Cultures

Huvec, MCF-7, and SH-SY5Y cell lines were acquired from the American Type Culture Collection and kept in Eagle’s minimum essential medium, supplemented with nonessential amino acids, 10% fetal bovine serum, penicillin (100 IU/mL), and streptomycin (100 µg/mL). The cell lines were cultivated in a 5% humidified CO_2_ atmosphere at 37 °C. Moreover, Huvec cells were cultured in Endothelial Basal Medium (EBM) (EGM™ SingleQuots^®^ CAMBREX, Walkersville, MD, USA) and SH-SY5Y cells were differentiated, before the treatment, with 10 μM of all-trans-retinoic acid for 5 days. When the cells reached 60% confluence, they were treated with PA or ALE 10 μg/mL for 24 h and the specific experiments were carried out.

### 3.3. Measurement of Cell Viability

Cell viability was assessed by a colorimetric test based on the use of 3-(4,5-Dimethylthiazol-2-yl)-2,5-diphenyltetrazolium bromide (MTT). The three cell lines were seeded in the number 80 × 10^3^ and grown into 96-well plates. After performing the treatments as described, the growth medium was replaced with a phenol-free medium containing an MTT solution (0.5 mg/mL). After 4 h of incubation, 100 μL of 10% SDS was added to solubilize the formazan crystals, and the optical density was measured at wavelengths of 540 and 690 nm. The instrument used was a spectrophotometric reader (X MARK Microplate Bio-Rad, Hercules, CA, USA).

### 3.4. Reducing Power Assay

To measure the reducing power of ALE and PA, a spectrophotometric assay was carried out, in which the Fe^3+^-Fe^2+^ transformation method was evaluated as reported above [41]. Ascorbic acid was used as a standard. Several aliquots of various concentrations of the standard (0.01–0.32 mg/mL) and our extracts (10 µg/mL) were added to 1.0 mL of deionized water. After mixing with 2.5 mL of phosphate buffer (pH 6.6) and 2.5 mL of ferricyanide potassium (1%), the samples were incubated at 50 °C in a bath for 20 min. Finally, 2.5 mL of trichloroacetic acid (10%) was added and the resulting solutions were centrifuged (3000 rpm for 10 min). Each top layer was mixed with 2.5 mL of distilled water and 0.5 mL of ferric chloride solution (0.1%, Sigma Aldrich) freshly prepared. The relative absorbance was measured spectrophotometrically at 700 nm. An increase in the absorbance of the reaction mixture has indicated an increase in reducing power. The instrument used was a spectrophotometric reader (X MARK, Hercules, CA, USA).

### 3.5. Ferrous Ion (Fe^2+^) Chelating Activity Assay

This assay measures the chelation of the iron ion and is used to evaluate the antioxidant potential. A total of 1 mL of sample was mixed with 1 mL of methanol and 0.1 mL of 2 mM FeCl_2_ and the reaction was started by adding 0.2 mL of 5 mM ferrozine. The resulting mixture was kept at room temperature for 10 min and its absorbance was measured at 562 nm. Ascorbic acid was used as a positive control.

### 3.6. ORAC Assay

The ORAC assay investigates the antioxidant capacity of ALE and PA by evaluating the transfer of hydrogen atoms and measuring the fluorescence loss over time of the fluorescein (used as a probe). Fluorescence is generated by the formation of peroxyl radicals, following spontaneous degradation of 2,2′-azobis-2-methylpropanimidamide, dihydrochloride (AAPH), which occurs at 37 °C. The peroxyl radical oxidizes the fluorescein and causes the gradual loss of the fluorescent signal. The use of antioxidants suppresses this reaction and inhibits signal loss. 6-Hydroxy-2,5,7,8-tetramTethylchroman-2-carboxylic acid (Trolox) inhibits fluorescence decay. Trolox was produced in PBS (pH = 7.0) at concentrations of 7.65, 15.25, 30.5, and 61 μg/mL, while extracts were used at a concentration of 10 μg/mL. The fluorescein fluorescent decay was evaluated using a microplate reader at wavelengths of 485 and 520 nm for excitation and emission, respectively. The measurements were taken in triplicate every 2 min for 1.5 h, and the data obtained from the fluorescence curves over time showed the average antioxidant efficacy of the samples. A regression equation was constructed by comparing the net area below the fluorescein decay curve and the Trolox concentration. The area under the curve has been calculated with the following equation:
i = 90
AUC = 1 + Σ f1/fo
i = 1

### 3.7. ROS Accumulation Measurement

The assay for measuring ROS is performed with the molecule fluorescein (H_2_DCF-DA), which easily penetrates cells and is cleaved by intracellular esterase to form H_2_DCF. This compound can no longer leave the cells and binds to the ROS present. H_2_DCF is transformed into DCF, which is highly fluorescent. The quantification of DCF is proportional to the content of ROS. Specifically, the cells were seeded in 96-well microplates with a density of 6 × 10^4^ cells/well, and the next day were treated with extracts, as described. After 24 h, the medium was replaced with fresh medium containing H_2_DCF-DA (25 μM) for 30′. At the end of the exposure time, cells were washed with PBS, and treated or not with H_2_O_2_ (150 μM, 30 min). Finally, cells were subjected to cytofluorimetric analysis (FACS Accury, Becton Dickinson, Franklin Lakes, NJ, USA).

### 3.8. Lipid Measurement: Spectrometric Quantification

The cell lipid content was measured using the dye Nile red, which binds closely to these biological molecules. Following exposure to ALE and PA, cells were incubated with 1 μg/mL of Nile red dye for 15 min. Subsequently, the cells were washed in PBS (pH = 7.4) and a lipid extraction was performed using the Bligh and Dyer method [41]. The lipid content of each sample was obtained by spectrometric reading (excitation/maximum emission ~552/636) and the results were interpolated with a straight line constructed with increasing concentrations of Nile Red. Values were normalized for cell DNA content (Bio-Rad kit, Milan, Italy). The instrument used was Multiskan^TM^ GO, Thermo Scientific™ (Milan, Italy).

### 3.9. Intracellular Calcium Measurements

Intracellular calcium was measured using the indicator Rhodamine 2 (Rhod 2, Molecular Probes, Eugene, OR, USA) which binds firmly to this ion (excitation/emission ~552/581). After treatment with extracts, cells were exposed to 5 μM of Rhod 2 for 1 h at 25 °C and protected from light. Subsequently, the cells were washed with PBS free of calcium and magnesium to prevent the entering of calcium from the outside. A first cytofluorimetric reading (which provided the basal concentration of calcium ion) was carried out, followed by exposure of the cells to thapsigargin (1 μM per 200 s). Tapsigargin is a non-competitive inhibitor of the Ca^2+^ ATPase on the endoplasmic reticulum (SERCA), which promotes calcium release from the organelle by increasing cytosolic calcium. Under these conditions, three cytofluorimetric readings were made. Finally, after 100 s, ethylene glycol-bis (β-aminoethyl ether)-N,N,N0,N0-tetraacetic acid (EGTA) 500 μM was added to chelate calcium [42,43]. It is important to point out that, before the readings, the cells were also treated with mitochondrial oxidative phosphorylation decoupling, carbonyl cyanide-4-(trifluoromethoxy)phenylhydrazone (FCCP, 1 μM), as well as with the inhibitor of the mitochondrial ATPase, oligomycin, (1 μg/mL). These compounds exclude the involvement of mitochondrial calcium. The cytofluorimeter used was an apparatus FACS Accury (Becton Dickinson, Milan, Italy).

### 3.10. Cell Lysis and Immunoblot Analysis

Cells were washed with cold PBS and lysed with a preheated buffer (at 80 °C) containing 50 mM Tris-HC (pH 6.8), 2% SDS, and a mixture of protease inhibitors. The lysates were immediately boiled for 2 min and the protein concentration was determined by the DCA protein test. Subsequently, 0.05% blue bromophenol, 10% glycerol, and 2% β-mercaptoethanol were added. Samples were boiled again and loaded into SDS polyacrylamide gels. After electrophoresis, the polypeptides were transferred to nitrocellulose filters, blocked with TTBS/milk (TBS 1%, Tween 20, and non-fat dry milk 5%), and then the antibodies were used to reveal their respective antigens. The primary antibodies were incubated overnight at 4 °C, followed by incubation with a horseradish peroxidase-conjugated secondary antibody for 1 h at room temperature. Blots were developed by chemiluminescence. The following primary antibodies were used: a rabbit Polyclonal anti-PERK antibody (GTX129275, GeneTex, San Antonio, TX, USA) at 1:1000 dilution, a rabbit Polyclonal anti-phospho-PERK antibody (29546-1-AP, Sigma Aldrich) at 1:500 dilution, a mouse Monoclonal anti-eIF2alpha antibody (D-3: sc-133132, Santa Cruz, Biotechnology, Dallas, TX, USA) at 1:1000 dilution, a rat Monoclonal Phospho-eIF2 alpha antibody (S52, Bio-techne, Milan, Italy) at 1:500 dilution, and a mouse Monoclonal Anti-β-Tubulin antibody (T8328, Sigma Aldrich) at 1:5000 dilution. Horseradish goat anti-mouse and anti-rabbit HRP-conjugated antibodies were used as a secondary antibody at 1:20,000 dilution. The blots were developed with ECL-PRIME reagent (Perkin Elmer, Monza, Italy).

### 3.11. Measurement of ATF-4 Expression Through Immuno-Cytofluorometry

The expression of transcription factor ATF4 was analyzed by immune-cytofluorimetry. Cells were collected in a growth medium in cytofluorometer tubes. The samples were treated with 10% BSA for 60 min at room temperature to block non-specific sites. Subsequently, the cells were incubated for 2 h at 37 °C with a mouse Monoclonal anti-ATF-4 antibody (CL594-60035, Thermo Fisher Scientific, Milan, Italy) at a dilution 1:200. At the end of the time required and following a wash with PBS to remove excess primary antibody, a secondary antibody conjugated with FITC was used, diluted in animal serum 5% for 1 h at room temperature. The required dilution was 1:400. After further washing in PBS, the reading at the cytofluorimeter was carried out. The cytofluorimeter used was an apparatus FACS Accury (Becton Dickinson).

### 3.12. Annexin V Staining

The cells, after being treated as described, are washed twice with cold PBS and resuspended in 1× Binding Buffer (Metabolic Activity/AnnexinV/Dead Cell Apoptosis Kit) at a concentration of 1 × 10^6^ cells/mL. One hundred microliters of suspension were mixed with 5 μL of FITC Annexin V (BD Biosciences, San Jose, CA, USA). The samples were gently vortexed and incubated for 15 min at 25 °C in the dark. Finally, 400 μL of 1× Binding Buffer and 5 μL of propidium iodate (PI) were added to each tube for 1 h, and the samples were analyzed by flow cytometry. The cytofluorimeter used was an apparatus FACS Accury (Becton Dickinson).

## 4. Conclusions

The results obtained enable us to answer the questions set out:(1)ALE and PA dissolve better in butanol, rather than methanol, ethanol, and propanol. Butanol has a higher number of -CH_2_ groups, which makes the solvent more hydrophobic and facilitates the dissolution of andrographolide contained in different amounts.(2)After ALE and PA were dissolved, they were subjected to a sonication cycle of 10′, responsible for better dissolution and greater effect. This additional treatment, used on *A. paniculata* extracts for the first time, has made a unique and innovative ALE and PA.(3)ALE and PA have shown an anti-tumoral effect on cancer cells, which is not present in healthy cells. This effect altered the physiology of the endoplasmic reticulum, which promoted cell apoptotic death. For this reason, *A. paniculata* could be used and exploited in pharmacological therapy against certain neoplasms. However, it should be remembered that the results were obtained in isolated cancer cells, which do not enjoy the homeostasis generated by a whole tissue or organ. In the organism, an anti-tumoral drug fails to distinguish between cancer and non-cancer cells, constituting the biggest limit of cancer therapy. It would be appropriate, therefore, to test ALE and PA in vivo or in clinical practice to increase information on the phytotherapeutic role of *A. paniculata*.

## Figures and Tables

**Figure 2 ijms-26-00344-f002:**
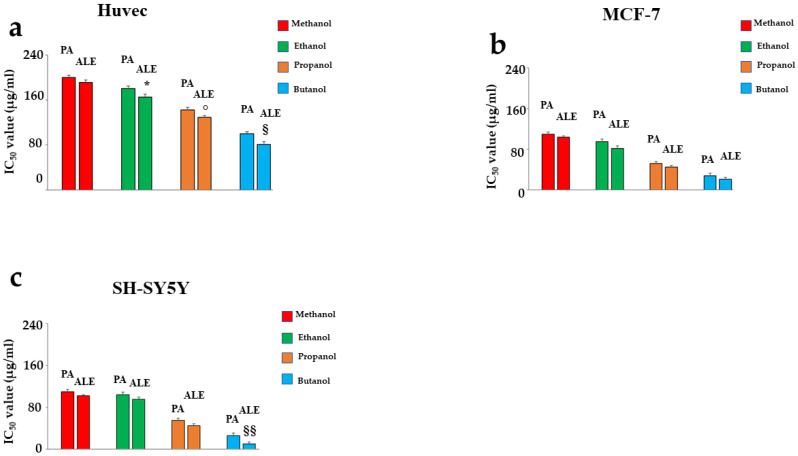
Comparison of IC_50_ of PA and ALE dissolved in several solvents. Figure 2 shows the variation of IC_50_ when PA and ALE are dissolved in different solvents. This analysis was conducted in Huvec, MCF-7, and SH-SY5Y cells as shown in panels (**a**–**c**), respectively. Three independent experiments were carried out, and the values are expressed as the mean ± SD. * denotes *p* < 0.05 vs. the respective PA; ° denotes *p* < 0.05 vs. the respective PA § denotes *p* < 0.05 vs. the respective PA; §§ denotes *p* < 0.01 vs. the respective PA. A Tukey–Kramer comparison test followed the analysis of Variance (ANOVA).

**Figure 3 ijms-26-00344-f003:**
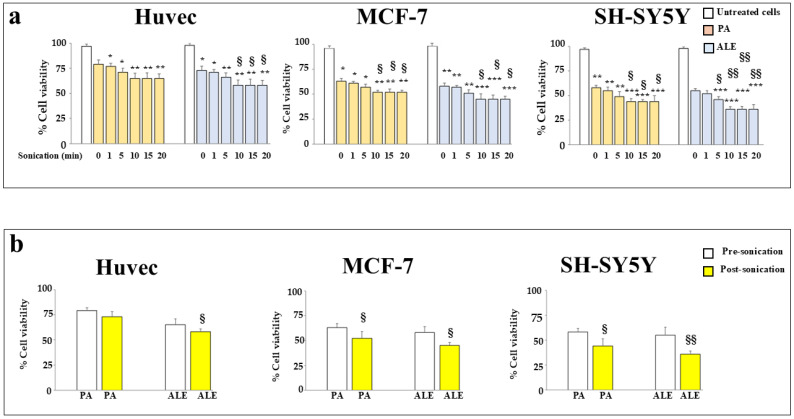
Effects of extracts following sonication. In Panel (**a**), cell lines were treated or untreated with ALE or PA, and extracts were subjected to a sonication process at different times. As can be seen in the legend of the figure, for each cell line the white histograms correspond to the viability of untreated cells, while the light pink and blue histograms correspond to the viability of cells treated with PA and ALE, respectively. Three independent experiments were carried out, and the values are expressed as the mean ± SD. * denotes *p* < 0.05 vs. untreated cells; ** denotes *p* < 0.01 vs. untreated cells; *** denotes *p* < 0.001 vs. untreated cells. § denotes *p* < 0.05 vs. the sonication time of 0′; §§ denotes *p* < 0.01 vs. the sonication time of 0′. A Tukey–Kramer comparison test followed the analysis of Variance (ANOVA). In Panel (**b**), differences in cell viability, obtained before and after sonication for 10′, were summarized. White histograms indicate cell viability obtained before the sonication of the extract, while yellow ones indicate cell viability obtained after the sonication of the extract. Three independent experiments were carried out, and the values are expressed as the mean ± SD. § denotes *p* < 0.05 vs. respective un-sonicated extract; §§ denotes *p* < 0.01 vs. respective un-sonicated extract.

**Figure 4 ijms-26-00344-f004:**
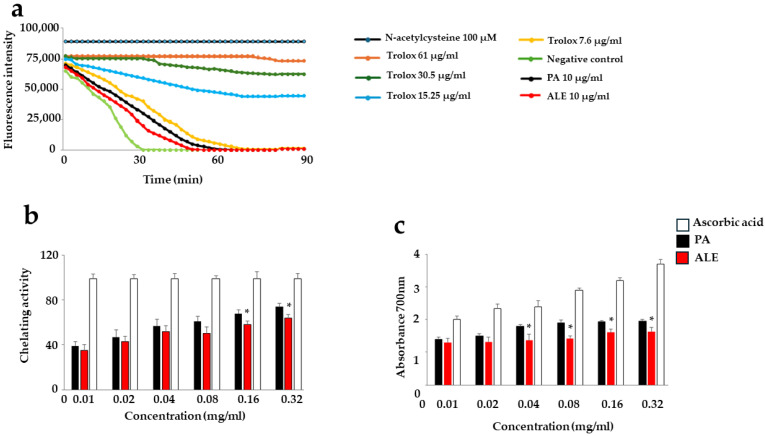
Panel (**a**) shows the antioxidant activity of PA and ALE obtained by the ORAC test. The curves corresponding to PA and ALE 10 μg/mL are represented in black and red, respectively. Three independent experiments were carried out and a representative experiment was shown. Panels (**b**,**c**) highlight the chelating and reducing properties of PA and ALE at increasing concentrations (0.01–0.32 mg/mL). Three independent experiments were carried out and the values are expressed as the mean ± SD. * denotes *p* < 0.05 vs. ALE. A Tukey–Kramer comparison test followed the analysis of Variance (ANOVA).

**Figure 5 ijms-26-00344-f005:**
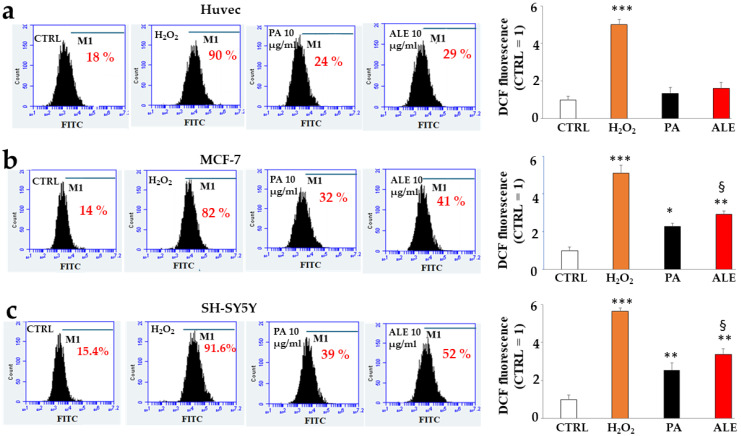
Production and accumulation of ROS. Panels (**a**–**c**) show the accumulation of ROS caused by PA and ALE in Huvec, MCF-7, and SH-SY5Y cells, respectively. The results were obtained from cytofluorimetric readings, and on the right of the plots, the respective quantifications are shown. Hydrogen peroxide was used as the positive control. Each box refers to a treatment as indicated: the *x*-axis represents the fluorescence of the fluorochrome FITC connected to our fluorescent probe, while the *y*-axis is relative to the number of cells we decided to acquire (10,000). At the top of each box, there is a marker (M1), which is arbitrarily drawn in the control and kept the same for all other samples. The part of the peak included in M1 is indicated by a numerical percentage. In the respective quantification, the percentages were compared. The control percentage is arbitrarily made equal to 1 and the other values are related to it. The control is represented by the white histogram, PA and ALE are highlighted by black and red histograms respectively; finally hydrogen peroxide is shown in orange. Three independent experiments were carried out, and a representative experiment was shown. The values are expressed as the mean ± standard deviation (sd). * denotes *p* < 0.05 vs. untreated cells; ** denotes *p* < 0.01 vs. untreated cells; *** denotes *p* < 0.001 vs. untreated cells. § denotes *p* < 0.05 vs. PA. A Tukey–Kramer comparison test followed the analysis of Variance (ANOVA).

**Figure 6 ijms-26-00344-f006:**
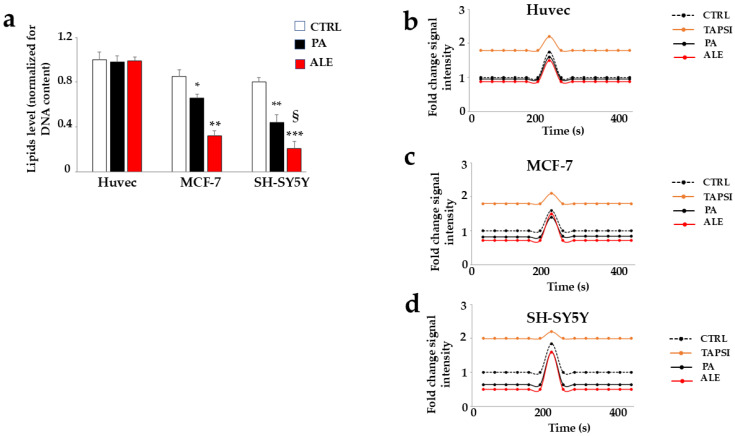
PA and ALE modify lipid and calcium concentrations. Panel (**a**) highlights the lipid concentrations resulting from treatment with PA and ALE in the selected cell lines. Panels (**b**–**d**) show the cytoplasmatic calcium concentration under the same experimental conditions. Thapsigargin, an endoplasmic reticulum stress inducer, has been used as a positive control. Three independent experiments were carried out. The values are expressed as the mean ± standard deviation (sd). * denotes *p* < 0.05 vs. untreated cells; ** denotes *p* < 0.01 vs. untreated cells; *** denotes *p* < 0.001 vs. untreated cells. § denotes *p* < 0.05 vs. PA. A Tukey–Kramer comparison test followed the analysis of Variance (ANOVA).

**Figure 7 ijms-26-00344-f007:**
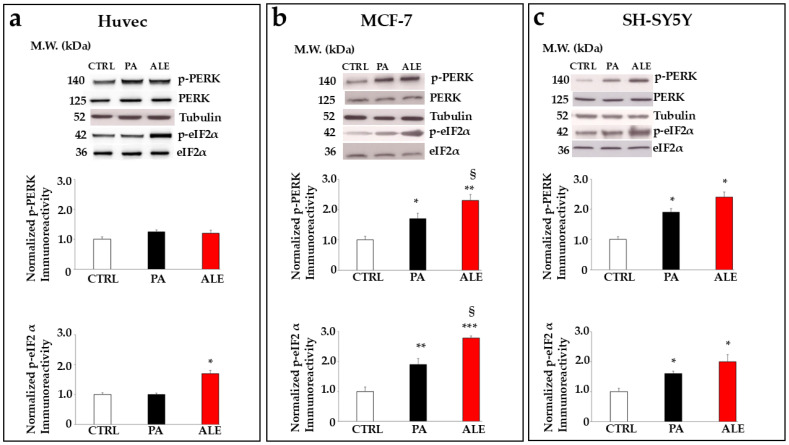
Involvement of UPR. Panels (**a**–**c**) show whether or not UPR is involved, following treatment with PA and ALE in Huvec, MCF-7, and SH-SY5Y, respectively. The studied UPR sensor was PERK, and its possible activation was justified by the modulation of the expression of its phosphorylated form (p-PERK) and by the phosphorylation of eIF2α. The results obtained were normalized for the housekeeping protein tubulin and one representative experiment was shown. The respective quantification is shown in the lower part of Figure 7. Three independent experiments were carried out. The values are expressed as the mean ± standard deviation (sd). * denotes *p* < 0.05 vs. untreated cells; ** denotes *p* < 0.01 vs. untreated cells; *** denotes *p* < 0.001 vs. untreated cells. § denotes *p* < 0.05 vs. PA. A Tukey–Kramer comparison test followed the analysis of Variance (ANOVA).

**Figure 8 ijms-26-00344-f008:**
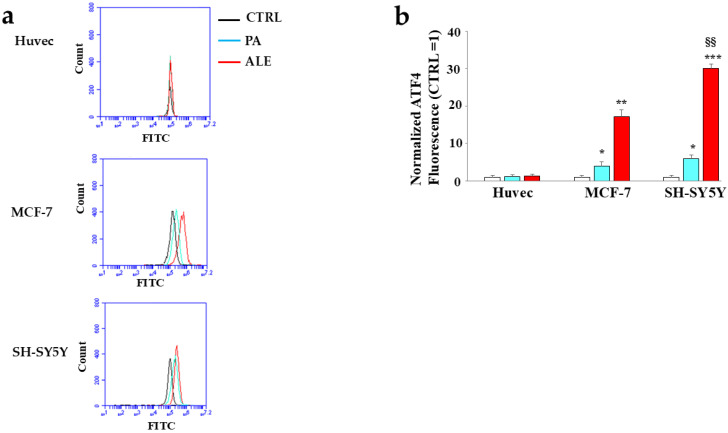
Expression of ATF4. Panel (**a**) shows the modulation of ATF4 expression in our experimental model. In every single box, the *x*-axis represents the fluorescence of fluorochrome FITC linked to our fluorescent probe and measured on a logarithmic scale; the *y*-axis is relative to the number of cells that we have decided to acquire (20,000). Panel (**b**) highlights the respective quantifications. On the plot, control is shown in black, while PA and ALE are shown in blue and red respectively. Three independent experiments were carried out. The values are expressed as the mean ± standard deviation (sd). * denotes *p* < 0.05 vs. untreated cells; ** denotes *p* < 0.01 vs. untreated cells; *** denotes *p* < 0.001 vs. untreated cells. §§ denotes *p* < 0.01 vs. PA. A Tukey–Kramer comparison test followed the analysis of Variance (ANOVA).

**Figure 9 ijms-26-00344-f009:**
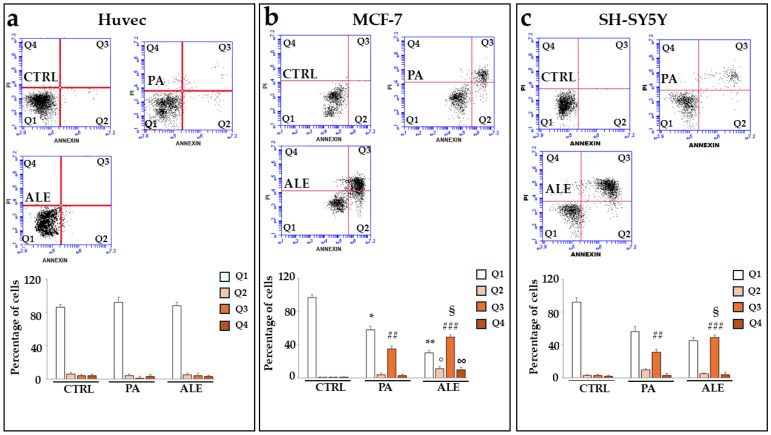
Treatment with PA and ALE: when is apoptosis present? To evaluate the effects of PA and ALE on the apoptotic process, cytofluorimetric experiments with annexin V/PI were conducted. Panels (**a**–**c**) represent these results obtained on Hucec, MCF-7, and SH-SY5Y cells, respectively. The upper part of the panels shows a representative experiment, while the lower part shows the respective quantification. Every box is divided into four quadrants (Q1, Q2, Q3, and Q4). Q1 refers to annexin V-negative/PI-negative cells (viable cells). Q2 refers to annexin V-positive/PI-negative cells (early apoptosis). Q3 refers to annexin V-positive/PI-positive cells (late apoptosis); Q4 refers to annexin V-negative/PI-positive cells (necrosis). Three independent experiments were carried out. The values are expressed as the mean ± standard deviation (sd). * denotes *p* < 0.05 vs. Q1 of Huvec cells; ** denotes *p* < 0.01 vs. Q1 of Huvec cells; ° denotes *p* < 0.05 vs. Q2 of Huvec cells; ## denotes *p* < 0.01 vs. Q3 of Huvec cells; ### denotes *p* < 0.001 vs. Q3 of Huvec cells; ∞ denotes *p* < 0.05 vs. Q4 of Huvec cells; § denotes *p* < 0.05 vs. PA. A Tukey–Kramer comparison test followed the analysis of Variance (ANOVA).

## Data Availability

Data is contained within the article and Appendix A.

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
