# Peer review of "An Innovative Method of Improving an Extract of *Andrographis paniculata* from Leaves: Its Anticancer Effect Involves the Cell Endoplasmic Reticulum"

_ijms, 2025, doi:10.3390/ijms26010344_

Round 1
Reviewer 1 Report
Comments and Suggestions for Authors
In the manuscript entitled “An innovative method of improving an extract of Andrographis 2 paniculata from leaves: its anticancer effect involves the cell 3 endoplasmic reticulum” the authors tested the effects of two extracts of Andrographis paniculata: one titrated at 49.7% andrographolide, obtained from the leaves of the plant, and the pure andrographolide at 99%. The extracts were tested on tumor lines (MCF7 and SH-SY5Y) and on a non-tumor line (Huvec), to evaluate their impact on cell proliferation. The authors show that the extracts were not toxic to Huvec cells, but damaged tumor cells, suggesting that cancer cells are more sensitive to the extracts than healthy cells. The observed damage was attributed to an alteration of the endoplasmic reticulum, which led to cell death by apoptosis. Therefore, A. paniculata showed antitumor potential and could be used in cancer therapy.
The work is well written and the experiments are well thought out and well done.
Minor points
1. I suggest that the authors test other cell lines as well. In fact, adding other tumor and non-tumor cell lines (e.g., lung, colon, or prostate cancer cells) could provide a more complete understanding of the effects of the extracts on different types of cancer and other healthy cells.
2. Furthermore, I consider it necessary also an in vivo experiment on animal models could be crucial to observe the effects of A. paniculata extracts in a whole organism, excluding potential variables related to cell culture. Furthermore, in vivo studies could reveal potential side effects or systemic toxicities not evident in in vitro tests.
3. A more detailed analysis of the PERK-eIF2α-ATF4 pathway may be necessary and enriched by the exploration of other UPR pathways (such as IRE1 and ATF6) and their interaction with other cellular signaling pathways, such as those involved in autophagy, cell cycle regulation and energy metabolism control. This would allow a more detailed analysis of the molecular pathways.
4. I also suggest to the authors molecular profiling studies. Use transcriptomic or proteomic techniques to investigate how treatment with ALE and PA affects gene and protein expression of tumor and non-tumor cells. This would allow to identify biomarkers of response or resistance.
5. Finally, add also pharmacokinetic and pharmacodynamic studies. In fact, studying the dose-response relationship and the pharmacokinetics of the extracts could provide crucial information for the optimization of the doses to be administered in a possible therapeutic treatment.
Author Response
Dear Reviewer,
Thank you for your valuable suggestions that will enhance my manuscript.
I suggest that the authors test other cell lines as well. In fact, adding other tumor and non-tumor cell lines (e.g., lung, colon, or prostate cancer cells) could provide a more complete understanding of the effects of the extracts on different types of cancer and other healthy cells.
- Furthermore, I consider it necessary also an in vivo experiment on animal models could be crucial to observe the effects of A. paniculata extracts in a whole organism, excluding potential variables related to cell culture. Furthermore, in vivo studies could reveal potential side effects or systemic toxicities not evident in in vitro tests.
In this study, attention was paid to the oxidative potential of the extracts and the discovery of the mechanism of action involved. Therefore, I believe that further investigation may be included in a later manuscript. The next step in this experimental work is to organise an in vitro study with additional non-cancer and cancer lines (lung and prostate). In parallel, we want to conduct an in vivo study on animal models affected by the same tumours. The use of additional cell lines may or not confirm the results obtained. In addition, the in vivo study may reveal any differences with cells useful for a possible future clinical trial. So, we imagine the first article as a precursor study of the effects of A. paniculata extracts and the next one as a completion and upgrading of the study.
- I also suggest to the authors molecular profiling studies. Use transcriptomic or proteomic techniques to investigate how treatment with ALE and PA affects gene and protein expression of tumor and non-tumor cells. This would allow to identify biomarkers of response or resistance.
This valuable suggestion could be included in the future manuscript.
- 3. A more detailed analysis of the PERK-eIF2α-ATF4 pathway may be necessary and enriched by the exploration of other UPR pathways (such as IRE1 and ATF6) and their interaction with other cellular signaling pathways, such as those involved in autophagy, cell cycle regulation and energy metabolism control. This would allow a more detailed analysis of the molecular pathways.
We have already studied the UPR branches of ATF6 and IRE1 as indicated in lines 298-299. Both are not involved as opposed to the PERK way. However, we will highlight it more understandably (Lines 306-308). The communication pathways involved in autophagy or cell cycle regulation were not studied because they would not be ready within five days, the time allowed for the revision of this manuscript. Furthermore, the type of cell death (necrosis and apoptosis) was assessed by annexin assay.
- 5. Finally, add also pharmacokinetic and pharmacodynamic studies. In fact, studying the dose-response relationship and the pharmacokinetics of the extracts could provide crucial information for the optimization of the doses to be administered in a possible therapeutic treatment.
The dose-response and time-response curves have been added. The explanations in the text can be highlighted in lines 171-175, while the data have been represented in the Supplementary Figure 2.
I wish you happy holidays and a happy new year.

Reviewer 2 Report
Comments and Suggestions for Authors
After carefully reading the manuscript entitled ‘An innovative method of improving an extract of Andrographis paniculata from leaves: its anticancer effect involves the cell endoplasmic reticulum ’, I noted several points that should be taken into account by the authors to improve the work.
1. What spectrophotometer was used in the iron ion (Fe2+) chelating activity test and Reducing Power Assay
2. please harmonise the writing of units, one is written ml and the other mL
3. Which ferric chloride was used in the method: Reducing Power Assay (line 411)
4.Were the analyses carried out in triplicate?
Having responded to the above points, recommends the paper for publication in the journal IJMS.
Author Response
Reviewer 2
Dear Reviewer,
Thank you for your valuable suggestions that will enhance my manuscript.
After carefully reading the manuscript entitled ‘An innovative method of improving an extract of Andrographis paniculata from leaves: its anticancer effect involves the cell endoplasmic reticulum ’, I noted several points that should be taken into account by the authors to improve the work.
- What spectrophotometer was used in the iron ion (Fe2+) chelating activity test and Reducing Power Assay
The instrument used was a spectrophotometric reader (X MARK Hercules, CA, USA). I have added this information in the text (Lines 422-423).
- please harmonise the writing of units, one is written ml and the other Ml
This change has been made to the entire manuscript.
- Which ferric chloride was used in the method: Reducing Power Assay (line 411)
Ferric chloride used is a solution purchased from Sigma Aldrich, as indicated in line 420.
4.Were the analyses carried out in triplicate?
Three independent experiments were carried out for every experiment, as indicated in the captures of images.
I wish you happy holidays and a happy new year.
